# A chemical toolbox for the study of bromodomains and epigenetic signaling

Qin Wu[1,2,3], David Heidenreich[1,4], Stanley Zhou [2,3], Suzanne Ackloo[1], Andreas Krämer[4,5], Kiran Nakka [6], Evelyne Lima-Fernandes [1,2,3], Genevieve Deblois[2,3], Shili Duan[2,3], Ravi N. Vellanki[2], Fengling Li[1], Masoud Vedadi[1], Jeffrey Dilworth [6], Mathieu Lupien [2,3], Paul E. Brennan[7], Cheryl H. Arrowsmith[1,2,3], Susanne Müller[4,5], Oleg Fedorov[7], Panagis Filippakopoulos[7] & Stefan Knapp [4,5,8]

Bromodomains (BRDs) are conserved protein interaction modules which recognize (read) acetyl-lysine modifications, however their role(s) in regulating cellular states and their potential as targets for the development of targeted treatment strategies is poorly understood. Here we present a set of 25 chemical probes, selective small molecule inhibitors, covering 29 human bromodomain targets. We comprehensively evaluate the selectivity of this probe-set using BROMOscan and demonstrate the utility of the set identifying roles of BRDs in cellular processes and potential translational applications. For instance, we discovered crosstalk between histone acetylation and the glycolytic pathway resulting in a vulnerability of breast cancer cell lines under conditions of glucose deprivation or GLUT1 inhibition to inhibition of BRPF2/3 BRDs. This chemical probe-set will serve as a resource for future applications in the discovery of new physiological roles of bromodomain proteins in normal and disease states, and as a toolset for bromodomain target validation.

[1] Structural Genomics Consortium, University of Toronto, Toronto, ON M5G 1L7, Canada. [2] Princess Margaret Cancer Centre, University Health Network, Toronto M5G 2M9 ON, Canada. [3] Department of Medical Biophysics, University of Toronto, Toronto M5G 2M9 ON, Canada. [4] Structural Genomics Consortium, Buchmann Institute for Life Sciences, Goethe-University Frankfurt, 60438 Frankfurt, Germany. [5] Institute of Pharmaceutical Chemistry, Goethe-University Frankfurt, 60438 Frankfurt, Germany. [6] Sprott Centre for Stem Cell Research, Regenerative Medicine Program, Ottawa Hospital Research Institute, Ottawa K1H 8L6 ON, Canada. [7] Target Discovery Institute and Structural Genomics Consortium, University of Oxford, Oxford OX3 7DQ, UK. [8] German Cancer Network (DKTK), Frankfurt/Mainz, 60438 Frankfurt, Germany. Correspondence and requests for materials should be addressed to C.H.A. (email: carrow@uhnresearch.ca) or to S.K. (email: knapp@pharmchem.uni-frankfurt.de)

Genetic and epigenetic variation, as well as environmental and lifestyle factors, work in concert to influence human health and disease. In recent years, the essential role of epigenetic modifications in regulating gene expression and cellular differentiation has emerged[1]. Apart from changes in DNA methylation, covalent post translational modifications (PTMs) of histones and other nuclear proteins define a complex language, the epigenetic code, which regulates chromatin structure and dynamics. Lysine acetylation (Kac) is a major epigenetic PTM occurring on histone proteins, which has been studied broadly. Kac has generally been associated with activation of transcription through opening of chromatin structure, although some recent studies have found some Kac marks to be responsible for the compaction of chromatin, protein stability, and the regulation of protein-protein interactions[2]. Disruption of histone acetylation patterns has been linked to the development of disease, which may occur through mutations that deregulate enzymes responsible for adding or removing these histone acetyl marks, as well as the protein interaction modules that recognize and interpret this important PTM[3].

Histone acetylation is a highly dynamic process that is regulated by histone acetyltransferases (HATs) and histone deacetylases (HDACs) that respectively write and erase acetylation marks. The complex pattern of acetylation marks is interpreted (read) by reader domains of the bromodomain (BRD) family of proteins. BRD-containing proteins are evolutionarily conserved and of substantial biological interest, as components of transcription factor and chromatin-modifying complexes and determinants of epigenetic memory[4]. There are 61 BRDs expressed in the human proteome, present in 46 diverse proteins. However, some atypical bromodomains, which lack essential residues, have little or no activity towards Kac-containing histone sequences and may recognize other epigenetic marks or unmodified peptide sequences, while canonical BRDs may also bind to complex patterns of modification around a central Kac site that often contain other PTMs [5–7].

The modular nature of many BRD-containing proteins, which typically harbor a number of diverse reader domains in addition to enzymatic functionalities and role(s) as scaffolds in large chromatin-modifying complexes, makes their functional study a challenging task. However, the development of highly selective inhibitors has provided versatile tools for functional studies on endogenous BRD-containing proteins, which can now be used to unravel the role of the epigenetic Kac-dependent reading process in chromatin biology as well as in the development of disease. This is exemplified by the development of highly potent inhibitors for BET (Bromo and extra-terminal; BRD2, BRD3, BRD4, BRDT) BRDs[8,9], which has led to numerous translational and functional studies on this subfamily of bromodomain proteins[10].

Chemical probes, small-molecule tool inhibitors with selectivity against similar proteins, have led to the validation of many disease targets, making seminal contributions to our understanding of complex cellular processes. However, chemical probes need to be highly selective, cell active, and therefore need to be comprehensively characterized in order to link observed phenotypic responses to targeted proteins. Unfortunately, selectivity and potency of tool compounds are often insufficient resulting in contradictory and erroneous results[11,12]. Following the disclosure of potent BET BRD inhibitors (BRDi), other members of the BRD family of interaction modules have been found to be highly druggable[13], resulting in the identification of chemical fragments that were subsequently developed into potent and selective chemical probes[14–20]. However, BRDs outside the BET family have not been found to be major regulators of primary transcription control, posing challenges for the discovery of functional roles of these conserved domains[21]. As a result, only a few studies have reported phenotypic consequences of inhibiting non-BET BRDs pointing to important roles in cellular differentiation[22,23].

Here, we characterize a comprehensive set of BRD chemical probes covering all subfamilies previously identified with good druggability scores[13]. Using a standardized commercial assay format (BROMOscan), based on a high-throughput binding assay originally developed to assess the selectivity of kinase inhibitors[24], we evaluate the selectivity of this BRD chemical probe-set and determine a total of 626 $K_D$ values on all detected interactions. We present here an overview of the binding modes of these inhibitors, resulting in the excellent selectivity of these chemical probes. To exemplify the use of this probe-set in biological systems, we further screen the collection on a cellular model of muscle differentiation identifying BET BRDs as major regulators in this context. Furthermore, systematic investigations of BRD inhibitors in triple-negative breast cancer (TNBC) cell lines have revealed an essential role for BRD inhibitors to target the metabolic vulnerability of TNBC, demonstrating their utility as a collection to uncover a previously unknown crosstalk between BRD components of the HBO1 HAT complex and cell metabolism. Together, our study provides a comprehensive structural and functional insight on BRD inhibitors, establishing a powerful resource for future mechanistic studies of this family of epigenetic reader domains, and underscoring the broad utility and immediate therapeutic potential of direct-acting inhibitors of human bromodomain proteins.

## Results

**A set of highly selective bromodomain chemical probes.** Chemical biology efforts of the past few years have led to the development of potent chemical probes for most BRD families (Fig. 1)[5]. The promyelocytic leukemia-SP100 nuclear bodies (family V), which harbor a PHD-BRD tandem reader cassette are a notable exception and some families are still insufficiently covered, including members of families VI and VII, which also have atypical and shallow Kac-binding pockets. In contrast, family I that contains the HATs PCAF and GCN5, as well as CECR2 and FALZ, is well covered by chemical probes. The dual PCAF/GCN5 chemical probe L-Moses showed good potency for these two highly related bromodomains ($K_D$ of 126 and 600 nM, respectively, determined by isothermal titration calorimetry (ITC))[25]. GSK4027 offers an alternative chemotype to antagonize the BRDs in these two targets with improved potency ($K_D$ 1.4 nM determined by BROMOscan for both BRDs)[26]. Early lead molecules for bromodomains of CECR2 and FALZ were discovered by screening a series of triazolophthalazines[27]. However, compounds of this series inhibited several BRD family members and exhibited poor solubility, limiting further development. NVS-CECR2-1 was the first potent chemical probe targeting CECR2 with good potency (80 nM, determined by ITC) and selectivity. An alternative probe molecule, GNE-886, has recently been published showing, however, some activity towards the BRDs of BRD9, BRD7, and TAF1/TAF1L[28].

To date, the BET BRDs (Family II) have had the greatest activity in inhibitor development, undoubtedly due to the strong and clinically relevant phenotypes observed for these compounds. This is an area that has rapidly evolved and has been previously reviewed in detail[10,29]. The first published Kac-competitive BRD inhibitors, which now have been widely used are the thienodiazepine (+)-JQ1 (henceforth, JQ1)[8] the related clinical compound OTX015[30], as well as the benzodiazepine iBET[9]. Inhibitors of this family show panBET activity primarily against the first BRD with slightly lower binding affinity towards the second BRD in BET proteins. More recently, antagonists featuring diverse Kac mimetics have been developed, including the isoxazole

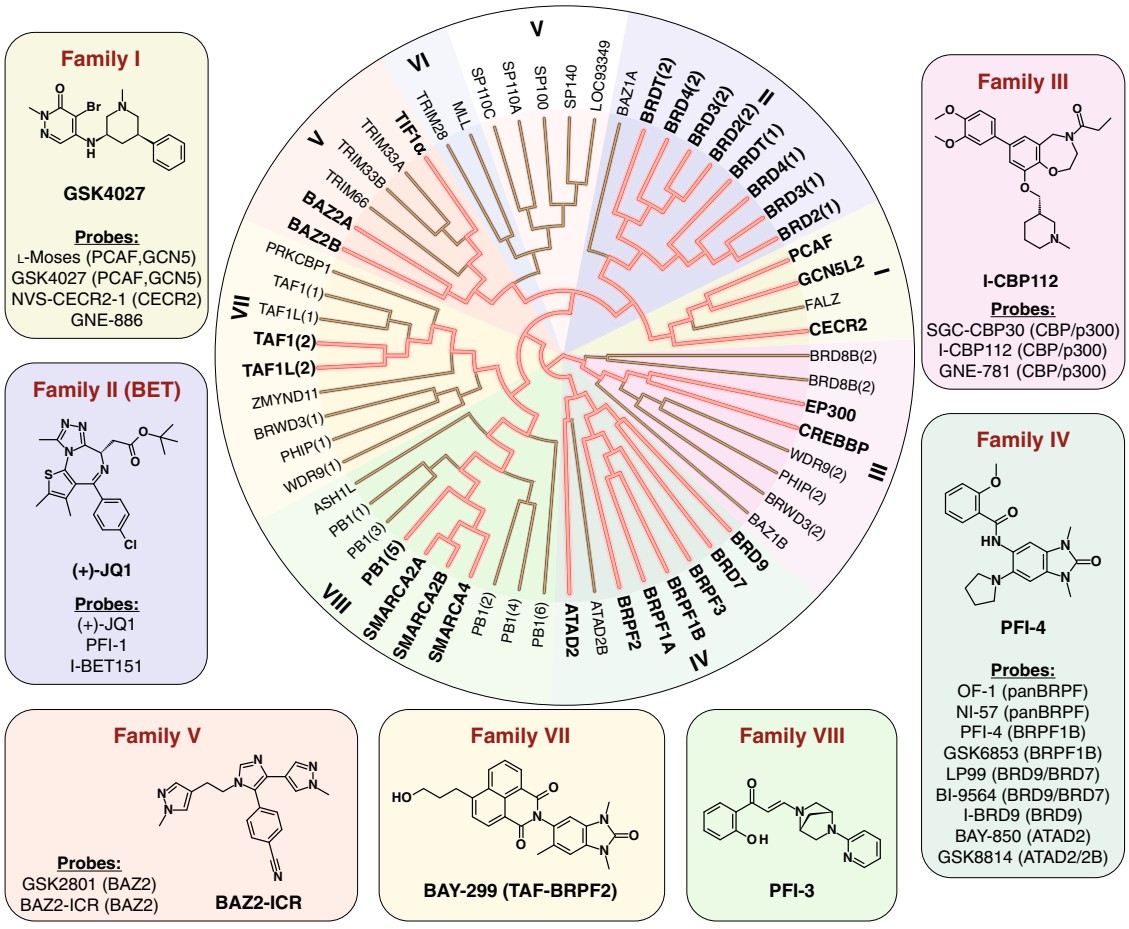

**Fig. 1** Chemical probes of the human bromodomain family. The set includes probes developed by our laboratory and a selection of additional inhibitors that are available. For each BRD family a single structural example of a chemical probe is shown. Additional probes are listed and a summary showing all chemical structures is included in Supplementary Table 1. BRD family members for which probes have been developed are highlighted in bold and by dark red lines in the dendrogram

I-BET151 (GSK1210151A)[31,32] and the tetrahydroquinazoline PFI-1[33]. Here we included in our probe-set JQ1, I-BET151, and PFI-1 as three structurally diverse and unencumbered chemical probes for BET proteins.

Family III contains BRDs present in the HATs p300 and CBP, as well as a number of diverse BRDs for which no potent inhibitors have been identified so far. The first inhibitor developed for CBP/p300, SGC-CBP30, exhibited potent activity for BRDs in these two HATs ($K_D$: 21 and 38 nM, respectively), retaining however significant BET affinity, which needs to be taken into account in cellular assays by using appropriate concentrations[18,34]. An alternative chemical probe is the benzoxazepine I-CBP112[23]. Recently, a highly potent antagonist, GNE-781, which has 650-fold selectivity over BRD4 for CBP/p300 became available[35].

Family IV contains BRDs participating in HAT scaffolding (BRPF1-3), and chromatin remodeling complexes (BRD7, BRD9, and ATAD2A/B). Several chemical probes target BRD7 and BRD9, including BI-9564[36] and LP-99[15], as well as I-BRD9, which has good selectivity towards BRD9 over BRD7[37]. The ATAD2 and ATAD2B BRDs have been targeted using the potent and selective antagonist GSK8814[38], while the allosteric BRD antagonist BAY-850 was developed as a specific probe against ATAD2[39]. BRPF BRD antagonists are well represented by the pan-BRPF chemical probes OF-1 and NI-57, and the two related BRPF1B-selective chemical probes PFI-4 and GSK6853[40,41]. In addition, BAY-299, a dual activity chemical probe for BRPF2 and

TAF1(2), has also been developed providing the only currently available chemical tool for family VII[42]. Similarly, compound 34, a dual activity antagonist of BRPF1B/2 and TRIM24 represents the only chemical tool currently developed for TRIM24[43].

Family VI is divided into the RING-type E3 ubiquitin transferase of the TRIM (tripartite motif-containing protein) family and BAZ2 (bromodomain adjacent to zinc-finger domain), which are components of chromatin remodeling complexes. As mentioned above, compound 34 offers proof of principle that BRDs within this family like TRIM24 are also druggable[43]. Two high affinity probes have also been developed against the BRDs of BAZ2A/B, GSK2801, and BAZ2-ICR[16,44]. Finally, BRDs in SMARCA2/4 (family VIII) (SWI/SNF related, matrix-associated, actin-dependent regulator of chromatin) and the scaffolding protein polybromo (PB1(5)) have been selectively targeted by PFI-3[22].

The chemical probe-set presented here comprises 25 tool compounds covering 29 (50%) of all human BRD proteins (Supplementary Table 1). Each chemical probe interferes with the binding of its respective BRD(s) acetyl-lysine target sequences, including the major acetylation sites described as BRD binding sites on histone proteins (Fig. 2)[5].

**Selectivity of bromodomain chemical probes.** Selectivity of a chemical probe is the most important feature of these key reagents. The BRD family contains 61 domains, many of them

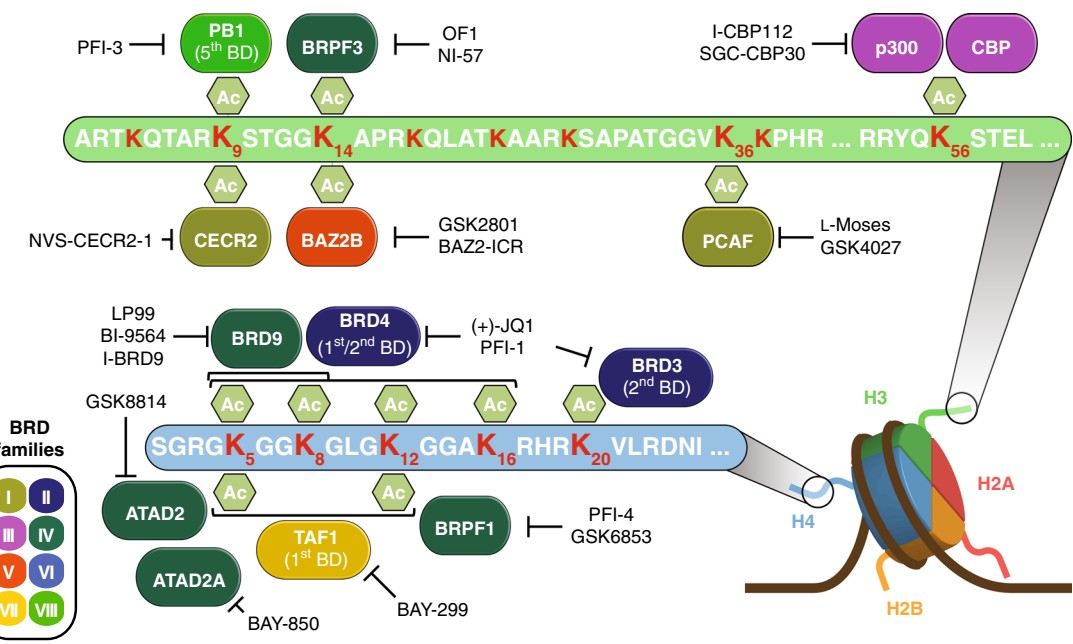

**Fig. 2** Schematic representation of interaction sites of current set of bromodomain (BRD) chemical probes. Major sites of lysine acetylation on histone H3 and H4 are highlighted in red

sharing significant sequence homology in the acetyl-lysine-binding domain and a common structural mechanism of Kac recognition. In particular, even partial inhibition of BET bromodomains by less selective BRD inhibitors has been linked to many phenotypic responses, thereby confounding attribution of the phenotype to a specific BRD protein[45]. The selectivity of chemical probes targeting diverse BRDs has been previously evaluated against a comprehensive set of recombinant human BRDs employing a temperature shift assay. This assay format makes use of the increase in the melting temperature ($\Delta T_m$) of a protein domain when complexed with a potent ligand[46]. However, intrinsic stability and other properties of proteins influence the magnitude of the observed temperature shift. Using bromosporine (BSP), a promiscuous BRD inhibitor[21], we evaluated selectivity screens against a panel of BRDs employing the BROMO*scan* ligand-binding assay, as well as ITC and thermal melt assays (Fig. 3). BROMO*scan* is a binding assay based on the well-known KINOME*scan* technology. This assay measures the binding of a DNA-tagged bromodomain to an immobilized BRD ligand. If an inhibitor is present, it will compete with the bromodomain binding to the immobilized ligand, resulting in reduction of a quantitative PCR (qPCR) signal in a dose-dependent manner. We used ITC as a standard for the accurate determination of binding constants, given its capacity to directly measure ligand binding in solution. All three assays resulted in comparable data (Fig. 3b) and we used this assay platform to determine affinities across 15 inhibitors of the probe-set (Fig. 3c). While correlation between ITC and BROMO*scan* data was excellent (Fig. 3d), some BRDs exhibited smaller than expected $\Delta T_m$ values based on their binding constants determined by ITC (Fig. 3e). In particular, BSP showed only modest $\Delta T_m$ against TAF1L(2) and BRD9 and a relatively large $\Delta T_m$ against CBP, compared to the directly determined ITC dissociating constants ($K_D$s). Encouraged by the accuracy of the BROMO*scan* assay, we screened 15 chemical BRD probes against 42 diverse bromodomains and determined a total of 626 dose–response curves (Supplementary data 1). In addition to the BRD probe-set, we included three closely related variants of chemical probes within our set, CBP30-298

and CBP30-383, which are closely related to SGC-CBP30, as well as PFI-3 D1, a close derivative of PFI-3 (Supplementary Fig. 1)[18,22,47]. However, while CBP30-related BET off-target effects were also apparent in the two additional CBP30 derivatives, the exclusive selectivity of PFI-3 towards SMARCA2/4 and PB1 was maintained in the derivative PFI-3 D1. Interestingly, the Kac mimetic salicylic acid head group of PFI-3 and its derivatives showed selectivity for this bromodomain subfamily. This striking observation has been rationalized by the unique binding mode of family VIII inhibitors that penetrated deeper into the Kac-binding site, leading to displacement of water molecules that are maintained in other BRD inhibitor complexes[48]. In summary, BROMO*scan* offers a robust platform for accurate $K_D$ determination of BRD inhibitors, and chemical probes screened here maintained at least a 30-fold selectivity window against BRDs in other families. In order to address selectivity outside the bromodomain family, we screened the chemical probes against 26 protein kinases, a target class that has been reported to be potently inhibited by some BRD inhibitors (Supplementary Table 2)[45,49]. In addition, the probe-set was also screened against other epigenetic modulators such as lysine and arginine methyl transferases, HATs, and methyl lysine reader domains (Supplementary Tables 3, 4, and 5). However, no significant activity of this probe-set was observed on any of these potential off-targets suggesting that this probe-set is selective toward bromodomains.

**BET inhibitors block muscle cell differentiation.** Chemical probes are versatile tools for studying the regulatory roles of the targeted proteins in complex cellular processes. Here we exemplify how the BRD probe-set can be used in a cellular system that has already been explored using the now well-studied BET inhibitors. Several BRD-containing proteins are known to be essential for different types of cellular differentiation, including myogenesis that we chose as a model system here[22,23,47]. During myogenesis, differentiating myoblasts fuse to form multinucleated myotubes/myofibers, a process that plays an important role in development and regeneration, and BET proteins have recently

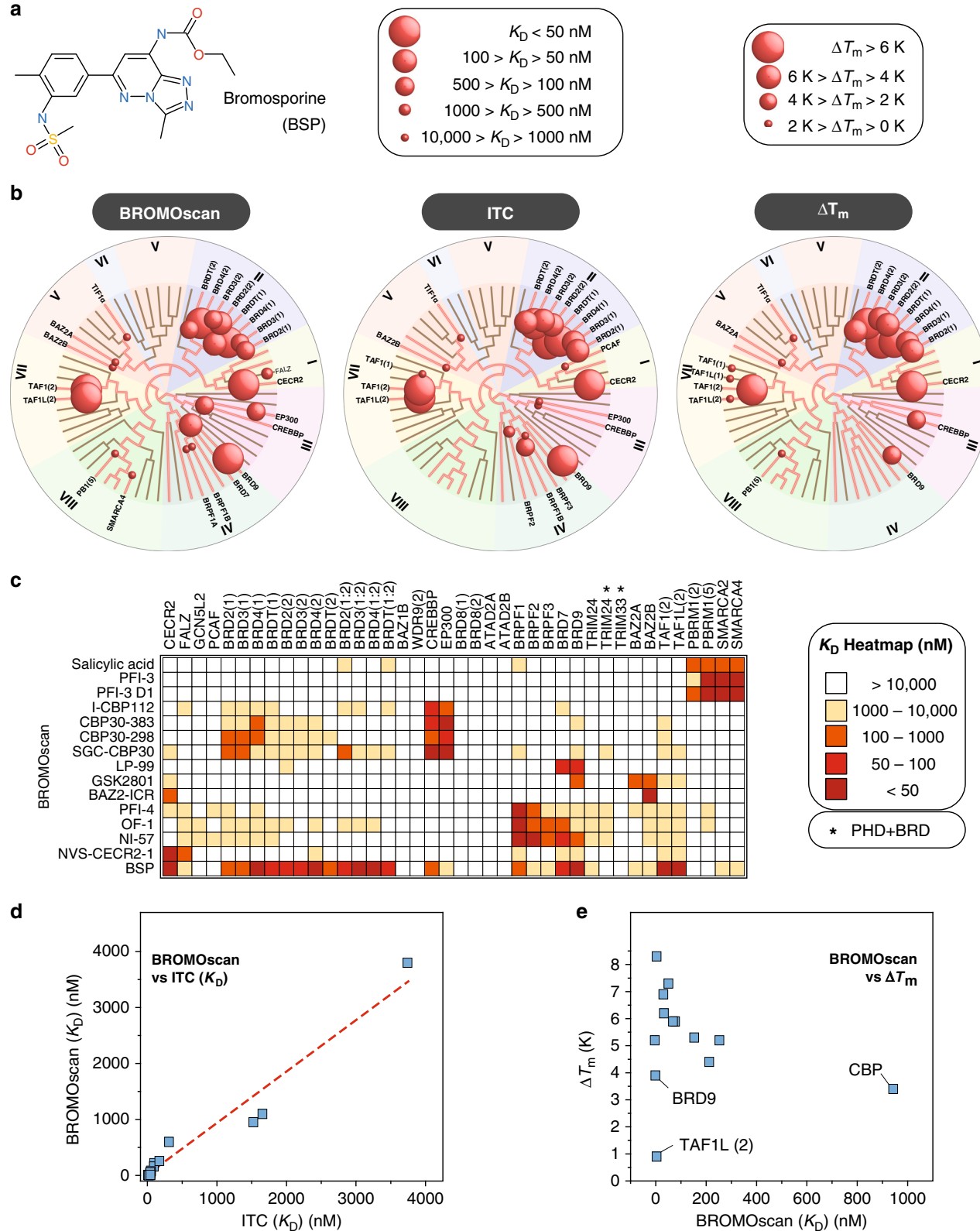

**Fig. 3** Selectivity of bromodomain chemical probes and assay comparison. **a** Structure of bromosporine (BSP). **b** Structural/phylogenetic dendrograms quantifying binding affinities of BSP to human BRDs measured by BROMO*scan* (left), ITC (middle) and $T_m$ assays (right). Affinities and $T_m$ shifts are mapped to the phylogenetic tree using spheres of variable sizes as indicated in the inset. Screened targets are annotated on the dendrograms. **c** Heatmap of measured BROMO*scan* $K_D$ values calculated from 10-data point dose–response curves. **d** Correlation of dissociation constants ($K_D$) measured by ITC and BROMO*scan*. **e** Correlation of $T_m$ shifts and dissociation constants ($K_D$) measured by BROMO*scan*

been implicated in this process[50]. Given the importance of muscle regeneration in human health, we were interested in determining whether BRD inhibition would modulate the ability of muscle progenitor cells to undergo terminal differentiation. To explore this, the muscle progenitor C2C12 cell line was cultured in conditions of low serum to initiate differentiation in the presence of diverse BRD inhibitors from the chemical probe-set described here. Strong inhibitory effects of myoblast differentiation were seen for the BET inhibitor JQ1 as well as the pan-BRD inhibitor BSP, but not for other BRD inhibitors (Figs. 4a, b). To gain insight into the transcriptional effects resulting from BRD inhibition, we performed gene-expression analysis (Fig. 4c). In agreement with our observations on myoblast differentiation, treatment with JQ1 and BSP resulted in significant changes in gene-expression levels. Importantly, promiscuous targeting of BRD proteins using BSP resulted in almost identical changes in gene expression compared to JQ1, strongly suggesting that the

inhibition of differentiation is due to BET BRDs and, among the targets tested, is not due to inhibition of other BRD-containing proteins. In agreement with this observation, expression changes resulting from targeting of other specific BRD inhibitors outside the BET family were negligible (Fig. 4c). Most significantly, differential expression analysis identified several anti-proliferative and anti-inflammatory genes down-regulated including proteins modulating interferon response, such as IFIT3 (interferon-induced protein with tetratricopeptide repeats 3), interferon-induced GTP hydrolases (GBPs), and USP18 (ubiquitin-specific peptidase 18) (Fig. 5a, b). Gene set enrichment analysis (GSEA) resulted in strong signatures for anti-inflammatory pathways and cell cycle regulators, as well as myogenesis (Fig. 5c; Supplementary Fig. 2). Gene ontology (GO) analysis corroborated these observations identifying enriched biological processes (BPs) relating to cell cycle and mitosis, as well as immune system processing and innate immune response (Fig. 5d, Supplementary

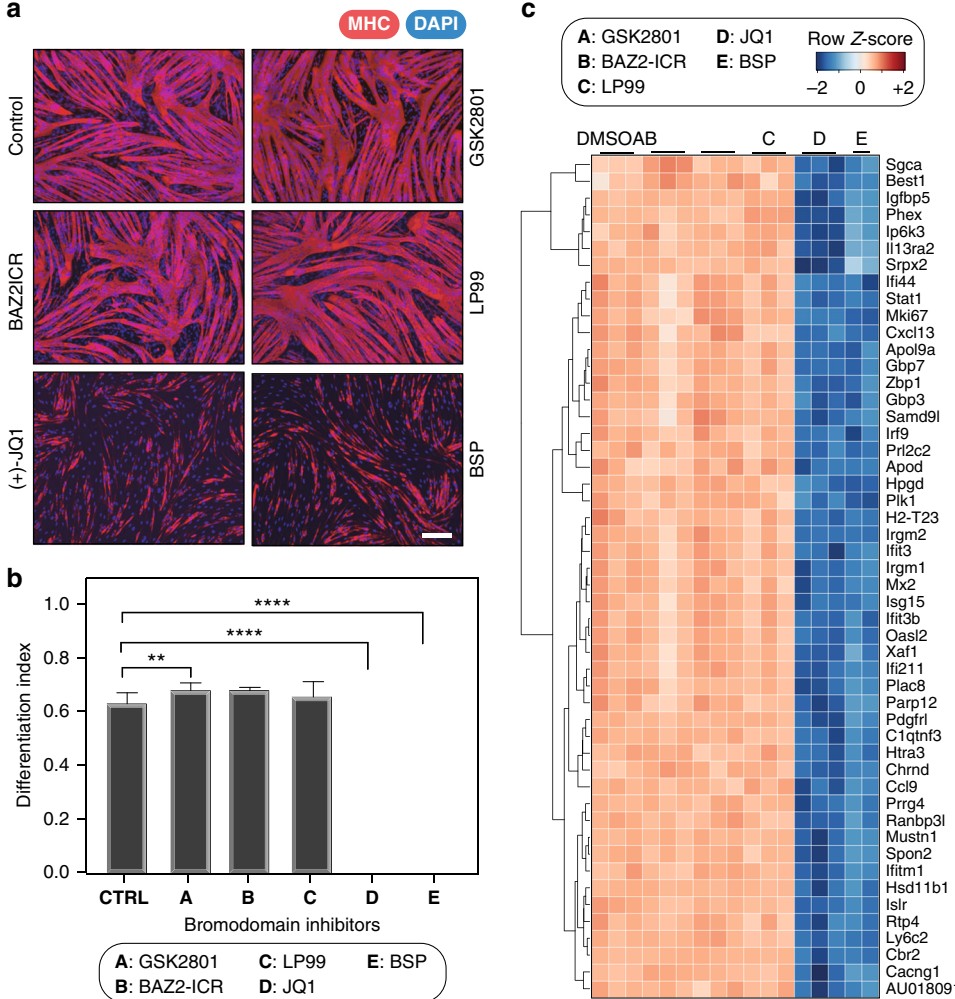

**Fig. 4** Influence of bromodomain inhibition on C2C12 myoblast differentiation. **a** Fluorescent images of myotubes cultured in differentiation media (Dulbecco's modified Eagle's medium (DMEM) containing 2% horse serum, 10 µgml$^{-1}$ insulin, and 10 µgml$^{-1}$ transferrin) in the presence or absence of bromodomain chemical probes. Cells were allowed to differentiate for 48 h before they were processed for immunofluorescence staining with α-myosin heavy-chain antibody (Red). Nuclei are stained blue with 4′,6-diamidino-2-phenylindole (DAPI). Images were acquired at ×10 using Zeiss Axio Observer Z1 microscope. **b** Quantitation of differentiated cells after inhibitor treatment. Post immunofluorescence, differentiation index was calculated by dividing the number of nuclei in myosin heavy-chain-expressing myotubes by the total number of nuclei per field. For JQ1 and BSP, the differentiation index declined significantly [****$p < 0.0001$] while treatment with GSK2801 slightly improved the differentiation index [**$p < 0.0053$]. Treatment with BAZ2-ICR ($p = 0.1773$) and LP99 ($p = 0.1959$) did not have significant impact on the differentiation index. P-values were calculated using two-tailed $t$ test and error bars represent standard deviation (s.d., $n = 3$). **c** Heatmap of the top 50 statistically significant genes that were differentially expressed (using the Benjamini-Hochberg adjusted $p$-value < 0.001) following 12 h treatment with specific BRD inhibitors

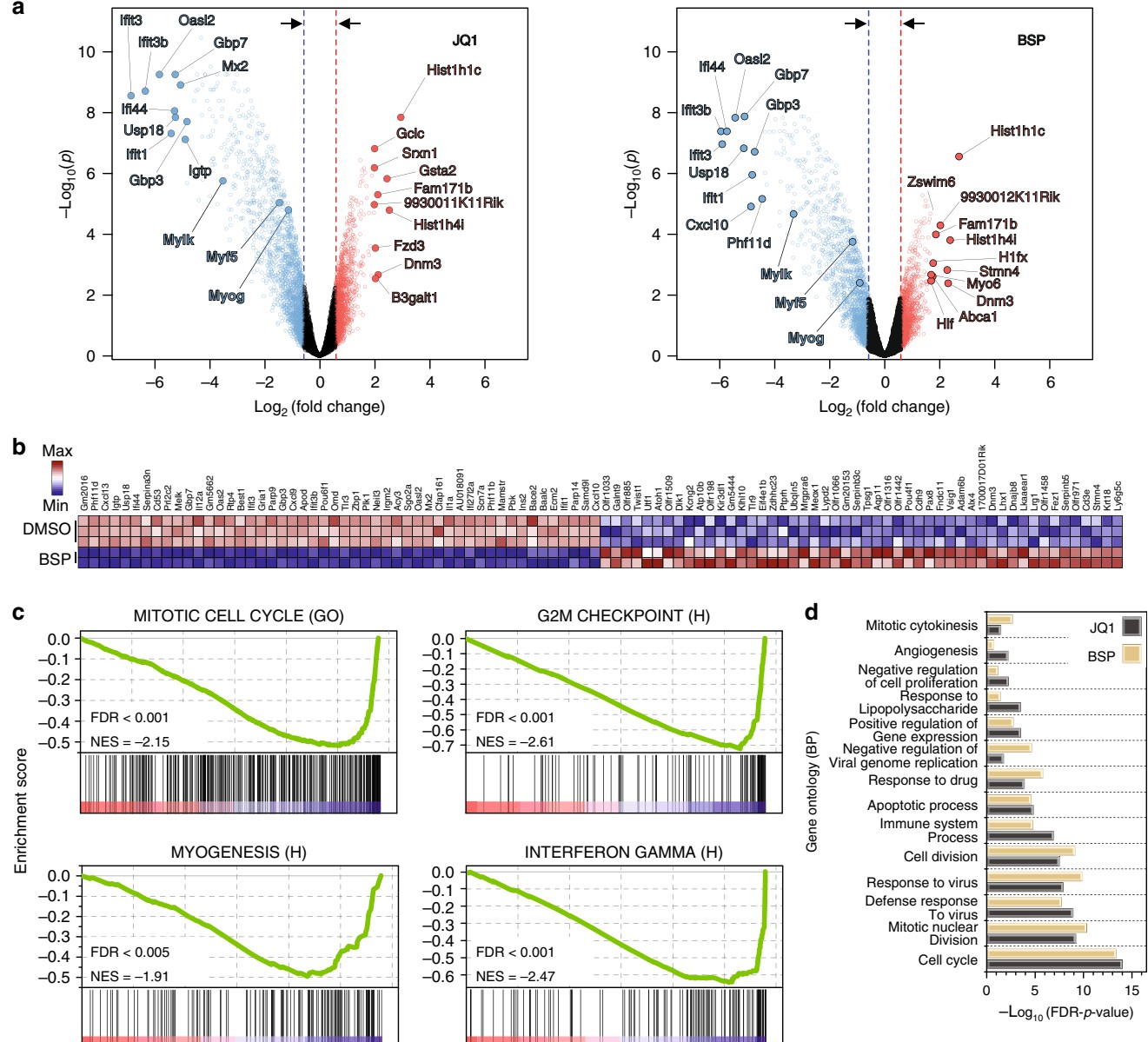

**Fig. 5** Transcriptional response to BSP and JQ1 in C2C12 myoblasts. **a** Volcano plot of differentially expressed genes following 12 h treatment with BSP (left) or JQ1 (right) in C2C12 myoblasts. The top 10 genes are sorted by their fold change and are highlighted and colored in red (up-regulated) or blue (down-regulated). **b** Heatmap of the top 50 up/down-regulated genes in C2C12 myoblasts following 12 h BSP treatment based on two-sided signal-to-noise ratio (SNR) score and $p < 0.05$ calculated by the Benjamini-Hochberg $t$ test. Dark blue indicates lowest expression; dark red indicates highest expression, with intermediate values represented by lighter shades, as indicated in the inset. Data are column normalized. **c** Gene set enrichment analysis (GSEA) demonstrating strong association with mitotic cell cycle (from the Gene Ontology (GO) MSigDB set, top left), G2M checkpoint (from hallmark MSigDB signatures, top-right), myogenesis (from hallmark MSigDB signatures, bottom-left), and interferon-γ (from hallmark MSigDB signatures, bottom-right) down-regulation signatures, following 12 h treatment of C2C12 cells with BSP. The plots show the running sum for the molecular signature database gene set within the C2C12/BSP data, including the maximum enrichment score and the leading edge subset of enriched genes. Normalized enrichment scores (NES) and false discovery rates (FDRs) are annotated in the insets. **d** GO enrichment (biological processes) for differentially expressed genes following 12 h treatment with JQ1 or BSP (calculated from differentially expressed genes with a Benjamini-Hochberg adjusted $p$-value < 0.001 and fold change >1.5)

Fig. 2). In particular, transcription of genes regulating expression of myosin light and heavy chains, as well as regulators of myosin, such as myosin light chain kinase (MLCK) was strongly suppressed. Importantly, we observed strong down-regulation of the muscle-specific basic-helix–loop–helix transcription factor myogenin (myogenic factor 4), a protein whose induction acts as a point-of-no-return in myogenesis by inducing cell cycle exit and activation of muscle-specific genes[51]. These observations suggest

therefore that transient inhibition of BET bromodomain-containing proteins may be a means to delaying myoblasts from undergoing terminal differentiation. Interestingly, we did not observe transcriptional regulation of MYC and its target genes (Supplementary Fig. 2), a gene-expression response that is frequently used as a marker for BET inhibition in cancer highlighting the context-dependent effect of BET inhibitors in different tissue types.

**BRD probes reveal a metabolic/epigenetic circuit in TNBC**. TNBC accounts for almost 20% of breast malignancies and is characterized by the lack of expression of the estrogen and progesterone receptors and absence of HER2 amplification[52]. Due to the lack of targeted therapies, patients with TNBC have a poor survival rate and a larger likelihood of distant recurrence and death within 5 years of diagnosis[53].

Recent studies showed that BET inhibitors such as JQ1 are effective against TNBC by specifically downregulating genes required for tumor growth and progression[54]. However, systematic investigations of the effects of other BRD inhibitors have not been evaluated in TNBC. To explore the potential anti-proliferation effects of BRD inhibitors, we profiled the viability of ten TNBC cell lines in the presence or absence of selected diverse BRD inhibitors of our BRD probe-set. In agreement with previous studies, BET antagonists, including JQ1 and PFI-1, display strong anti-proliferative effects on all the TNBC cell lines (Fig. 6a), likely due to BETi effects on super-enhancer-dependent transcription[55]. However, no significant growth inhibitory effects were observed for the remaining tested members of this BRD probe-set.

The use of synergistic drug combinations is an increasingly important concept for the development of new cancer treatment strategies. We were therefore interested if bromodomain targets that are inhibited by the current BRD probe-set would act synergistically with inhibitors that target other key drivers of tumor growth in TNBC. Metabolism is an important modulator of tumor growth and it can directly impact cellular epigenetic landscapes and alter responses to chemotherapeutics. In particular, acetylation of histones relies on the availability of the universal acetyl donor metabolite acetyl-CoA, which is biosynthesized by breakdown of carbohydrates through the glycolytic pathway. Many TNBC cell lines display a classical Warburg metabolism with up-regulated glucose uptake to fuel their bioenergetics and biosynthetic demands[56]. We confirmed that the set of 10 TNBC cell lines investigated in this study had a range of metabolically distinct states and variable global levels of histone acetylation (Supplementary Fig. 3a–c). Furthermore, compared to other breast cancer subtypes, TNBC cell lines have a higher glycolytic gene-expression signature, especially for glucose transporter I (GLUT1) expression (Fig. 6b), and thus tend to be more sensitive to glucose depletion[57]. Accordingly, this led us to investigate whether disruption of glycolysis in TNBC cell lines can give rise to epigenetic vulnerabilities to BRD inhibitors.

For this we chose the selective GLUT1 inhibitor BAY-876[58]. We first confirmed that exposure to BAY-876 inhibits glucose uptake in TNBC cell lines and then assessed the impact of its treatment on metabolites related to glycolysis and relevant to histone acetylation (Fig. 6c, Supplementary Fig. 3d). Acetyl-CoA is at the crossroads of glycolysis and TCA cycles and is the cofactor for HATs[59]. Upon treatment with BAY-876, a decrease in absolute acetyl-CoA level was observed (Fig. 6d). Another essential player involved in metabolism and acetylation is the $NAD^+/NADH$ ratio, which is closely associated with energy status in cell and is thought to positively regulate the activity of sirtuins[60]. We observed an increase in the $NAD^+/NADH$ ratio that may lead to increased sirtuin activity, possibly also contributing to histone hypoacetylation (Fig. 6e). Having observed changes in metabolites linked to lysine acetylation in response to BAY-876, we next investigated whether there are corresponding changes in histone acetylation levels. Interestingly, we detected a reduction in the global levels of acetylated histone H3 (ac-H3), but not in the global levels of acetylated histone H4 (ac-H4), in response to BAY-876 treatment (Fig. 6f). These results demonstrate that manipulating metabolic flux by inhibiting glucose uptake can specifically impact the acetylation on individual histones.

We next assessed whether altered histone acetylation induced by BAY-876 treatment could induce sensitivity to BRDi in TNBC cell lines. We performed a combinatorial screen on three TNBC lines with distinct glycolytic rates (Supplementary Fig. 3c). At a concentration of 3 μM, BAY-876 treatment alone had no effect on these three TNBC lines (Fig. 6g), but interestingly, in combination with the chemical probes PFI-1, OF-1, or I-BRD9 (3 μM each), we observed a significant decrease in viability (Fig. 6h). The activity of OF-1 was of particular interest due to the strong synergistic effect across all three cell lines, whereas JQ1 and I-BRD9 combinations with BAY-876 showed efficacy in only one of the three tested TNBC lines (Fig. 6h).

**OF-1 acts synergistically with GLUT1 inhibition**. To better understand the mechanism of the BAY-876/OF-1 combination, we determined the half-maximal inhibitory concentration ($IC_{50}$) values for OF-1 treatment in all three cell lines. In the presence of 3 μM BAY-876, we observed $IC_{50}$ values for OF-1 in the range of 0.3–2 μM (Fig. 7a). An increase of OF-1 sensitivity was also observed in response to glucose deprivation mimicking the effect of BAY-876 treatment (Fig. 7b). The combinatorial effect of BAY-876 and OF-1 was more than additive because the BAY-876 concentration we used in this assay has no overt effect on cell growth based on colony formation assay (Supplementary Fig. 3e). We next examined whether the observed synergy is due to the induction of apoptosis by the combination. Indeed, apoptosis markers such as caspase-3/7 were induced at a significantly higher level in combination-treated cells compared with cells treated with either OF-1 or BAY-876 alone (Supplementary Fig. 3f). Thus, we conclude that OF-1 and BAY-876 are synergistic in suppressing the growth of TNBCs by inducing apoptotic cell death.

OF-1 inhibits Kac binding of the BRDs of BRPF1, BRPF2, and BRPF3. In order to deconvolute which BRD proteins are responsible for the observed phenotype with BAY-876, we compared the combinational effect on TNBC cell viability with another potent pan-BRPF inhibitor NI-57, and the selective BRPF1 inhibitor PFI-4. Co-treatment with BAY-876 and OF-1 led to the strongest reduction of cell viability, whereas other combinations were less effective (Fig. 7c). Although NI-57 has higher binding affinity toward BRPF family proteins in vitro compared to OF-1, its lower solubility and 3-fold lower predicted cell permeability might affect its cellular activity, potentially explaining why no synergy was observed with BAY-876 (Supplementary Fig. 3g). Moreover, no significant effect was observed in the PFI-4 and JQ1 combination, which excludes the role of BRPF1 and potential contribution of BET off-target activity to the observed synergistic effect (Supplementary Fig. 3h–i). In agreement with these results, small interfering RNA (siRNA) knockdown of BRPF1 did not change the cell sensitivity to BAY-876 treatment (Fig. 7d–f). Notably, compared to the single BRPF2 or BRPF3 knockdown, the cells with dual BRPF2/3 knockdown are more sensitive to BAY-876 treatment (Fig. 7f). Together, these results demonstrate that the observed synergy effects are due to the inhibition of BRPF2 and/or BRPF3 by OF-1.

BRPF2 and BRPF3 are components of the HBO1 (KAT7) acetyltransferase complex, while BRPF1 preferentially participates in the MOZ/MOF complex[61,62]. Furthermore, BRPF2/3/HBO1 complexes were also shown to be important and specific for the HAT activity toward H3K14, whereas the BRPF1 complexes have high specificity on H4 acetylation marks[63]. To explore whether inhibition of BRPF2 or/and BRPF3 has any effect on histone acetylation, we measured the H3K14ac level upon treatment with BRPF BRD chemical probes, or in BRPF-knockdown cell lines. Compared to NI-57 and PFI-4, the acetylation of H3K14 was

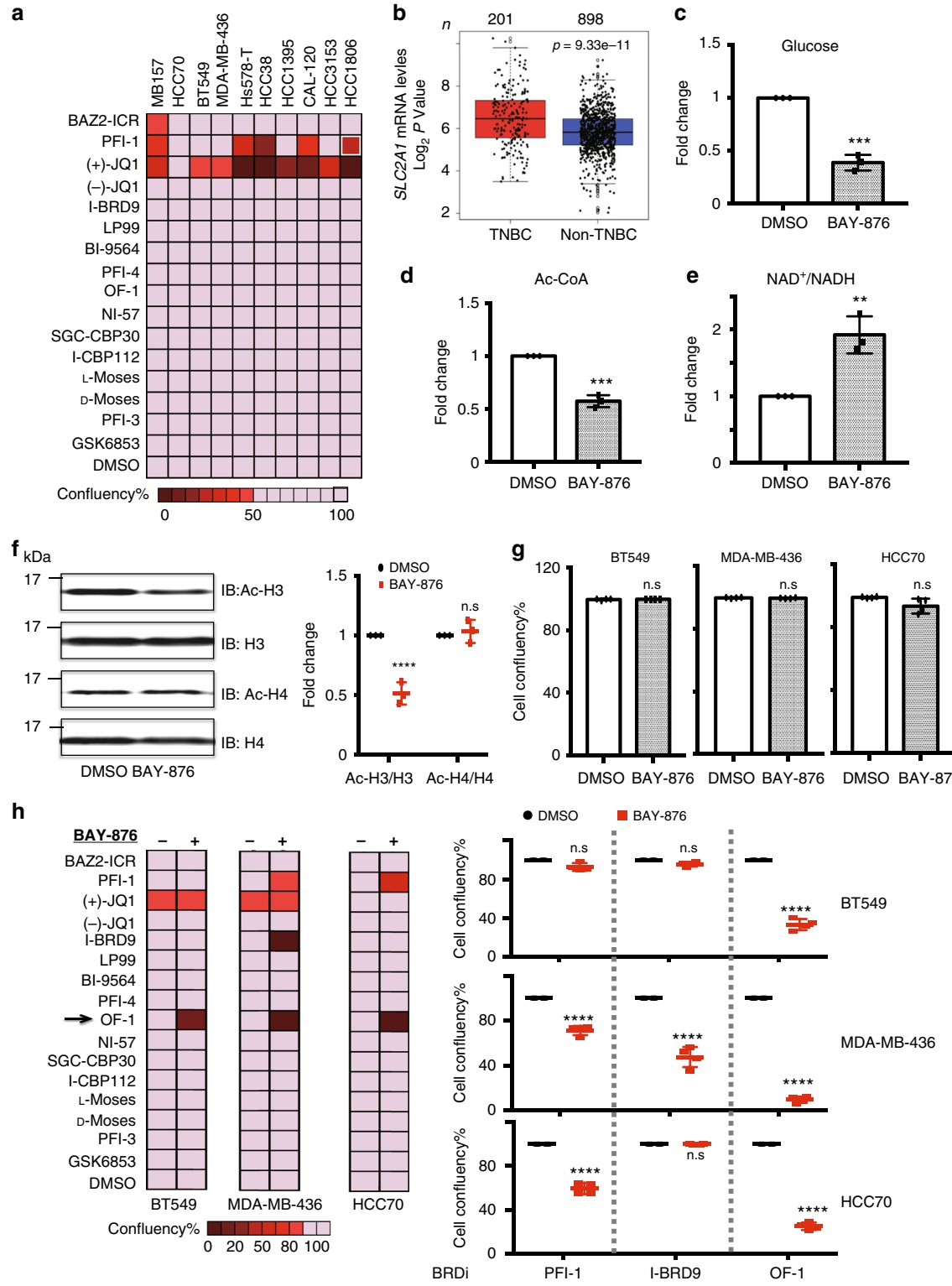

significantly decreased in the presence of OF-1 (Fig. 7g, Supplementary Fig. 3j). Likewise, dual BRPF2/3 knockdown displayed the strongest reduction of H3K14ac compared to BRPF2 or BRPF3 knockdown alone (Fig. 7h, Supplementary Fig. 3k). Taken together, these data are consistent with a model in which inhibition of glucose uptake by BAY-876 and antagonism of HBO1 subunits BRPF2/3 by OF-1 both converge on the same histone marks, leading to synergistic crosstalk between metabolic and epigenetic pathways (Fig. 7i).

## Discussion

Recent effort by our laboratories and others established a comprehensive set of epigenetic probe molecules for selective targeting of acetyl-lysine-dependent reader domains. We believe that this is a significant achievement considering that apart from a number of fragment-like small molecules, no BRD had been targeted before the first potent BET inhibitors were disclosed in 2010[8,9]. In particular, BET inhibitors had a major impact on basic and translation research as demonstrated by the large number of

**Fig. 6** BRD inhibitors leverage metabolic adaptations induced by glucose transporter I (GLUT1) inhibition in TNBC. **a** BRD inhibitor screening across ten TNBC cell lines. Cells were treated with indicated BRD inhibitors at 3 µM for 7 days. Confluency was measured using an IncuCyte ZOOM live cell imaging device. Data shown are mean ± s.d. of n = 4 independent cell culture grown and treated cells. A two-sided Student's t test was used to derive the p-values. **b** SLC2A1 gene expression in the The Cancer Genome Atlas (TCGA) breast datasets. The cohorts were divided into TNBCs (red) and non-TNBCs (blue) according to PAM50 classification. Gene expression is reported as median-centered expression $\log_2$ values. The number of patients (n) per group is indicated. P-values were determined using a Wilcoxon's rank-sum test. **c** Glucose uptake in MDA-MB-436 cells in response to BAY-876 treatment relative to vehicle. MDA-MB-436 cells were treated with dimethyl sulfoxide (DMSO) or 3 µM BAY-876 for 5 days. Graph indicates mean, error bars denote s.d. from three independent assays and p-value was computed using the Benjamini-Hochberg t test; ***p < 0.001. **d** Effects of BAY-876 treatment on the intracellular acetyl-CoA level. Graph indicates mean, error bars denote s.d. from three independent assays and p-value was computed using the Benjamini-Hochberg t test; ***p < 0.001. **e** Effects of BAY-876 treatment on intracellular $NAD^+/NADH$ level. Graph indicates the mean, error bars denote s.d. from three independent assays and p-value computed using the Benjamini-Hochberg t test; **p < 0.01. **f** Immunoblot analysis of H3 and H4 acetylation in MDA-MB-436 cells before and after BAY-876 treatment. P-value computed using the Benjamini-Hochberg t test; ***p < 0.001. **g** Cell growth effects of BAY-876 treatment on three representative cell lines. n.s. not significant. **h** Combinatorial screening of BRD inhibitors with or without 3 µM BAY-876 across ten TNBC cell lines. Cell confluency was obtained from the endpoint Incucyte scanning. (Left) Heatmap of the combinatorial screening results; (Right) Cell confluency after treatment with three potential BRDi candidates at 3 µM in the presence or absence of 3 µM BAY-876 in three representative cell lines. Graph indicates mean, error bars denote s.d. from three independent assays and p-values were computed using the Benjamini-Hochberg t test; **p < 0.01; ***p < 0.001. Raw data images are available in Supplementary Fig. 4

research papers that have been published using these reagents and more than 20 clinical trials that are currently registered[10]. However, BRD inhibitors outside the BET family have not been validated as potential drug targets. Here we provide data showing that inhibition of BRPF BRDs in combination with selective inhibitors of glucose transport might be beneficial for the treatment of TNBCs. Earlier studies demonstrated also synergies of BRD inhibitors with other drugs, such as CBP/p300 inhibitors acting synergistically with BET inhibitors as well as cytotoxic agents and dexamethasone in leukemia[23]. In addition, BET inhibitors showed synergy in cancer models in combination with HDAC inhibitors[64,65]. The combination of different inhibitors might also be important in overcoming drug resistance, which has been observed in cells treated with BET inhibitors[66]. The reported surprising dual activity of kinase and BET inhibitors suggests that potent activity for both bromodomain and kinases could be designed into a single inhibitor[45,49,67].

The profiling data provided here offers a comparison of inhibitor potency and selectivity across the BRD family. We found good correlation of BROMOscan assay data with $K_D$s determined in solution by ITC, whereas the magnitude of $T_m$ shifts across the BRD family may vary depending on the intrinsic stability of each BRD. However, as an analytical tool the $T_m$ shift assay provides a good platform for assessment of selectivity when hits are carefully followed up using orthogonal binding assays. Some probe compounds were exclusively selective, while others, such as the CBP/p300 probes I-CBP112 and CBP30, showed significant BET activity (Supplementary Table 2). Thus, care should be taken when these probes are used in cellular assays. We recommend that probe concentration not higher than 3 µM for I-CBP112 and 2.5 µM for CBP30 are used in cell-based assays and that BET inhibitors are included as controls.

Even though the coverage of the bromodomain family with chemical probes is now quite good, there are still a number of BRDs for which no selective or even non-selective inhibitors are available. Many of these BRDs have unusual Kac-binding sites, for instance, in some BRDs the conserved Asn is replaced by Ser, Thr, and Tyr residues[5]. No specific Kac-containing sequences have been reported binding to these BRDs limiting the development of Kac-competitive assays. Some of these BRDs may also not recognize Kac-containing sequences at all. Other BRDs have less druggable binding sites making the development of high-affinity chemical probes challenging. There are now also structurally diverse Kac-binding domains called YEATs domains that have recently been targeted by small molecules and fragments[68,69]. It is therefore likely that the arsenal of chemical

probes for these reader domains will continue to grow in the future. New chemical probes and associated data will therefore be published on our web-based database (https://www.thesgc.org/chemical-probes).

Most bromodomain-containing targets are complex multi-subunit-containing molecules, which also contain histone- and chromatin-interacting proteins. For some BRD-containing proteins, such as BET proteins, chemical antagonism of Kac binding is sufficient to displace the target protein from its intended chromatin loci. In other cases, such as for p300/CBP- and SMARCA2/4-containing complexes, it appears that BRD antagonism is insufficient to displace the entire complex from chromatin[22,23]. Thus, BRDi targeting complex chromatin proteins are not likely to always replicate genetic knockdown studies of the full-length protein[22,70]. We believe that this chemical probe toolset will be an excellent resource for understanding the role of specifically targeted BRDs within larger chromatin complexes and will likely reveal novel opportunities for translational research projects.

## Methods

**Cell culture**. Human TNBCs were obtained from the American Type Culture Collection (ATCC). Cells were cultured in media recommended by the provider, their identity was confirmed by short tandem repeat analysis, and they were regularly tested for mycoplasma. MB157 (ATCC CRL-7721™), MDA-MB-436 (ATCC HTB-130™), Hs578-T (ATCC HTB-126™), and CAL-120 (RRID:CVCL_1104) cells are grown in Dulbecco's modified Eagle's medium (DMEM) supplemented with 10% fetal bovine serum (FBS), penicillin (100 Uml$^{-1}$), and streptomycin (100 µgml$^{-1}$). HCC70 (ATCC CRL-2315™), HCC1806 (ATCC CRL-2335™), HCC1143 (ATCC CRL-2321™), HCC3153 (RRID:CVCL3377), and BT549 (ATCC HTB-122™) cells are grown in RPMI supplemented with 10% FBS, penicillin (100 Uml$^{-1}$), and streptomycin (100 µgml$^{-1}$).

**High-throughput screening of bromodomain inhibitors in TNBC panel**. TNBC cell lines were seeded in sterile, transparent 384-well plates at 500 cells per well in 50 µl media. Chemical probes were delivered in dimethyl sulfoxide (DMSO). Cell confluency at 7 days was evaluated using an Incucyte ZOOM live cell imaging device (Essen Bioscence) and analyzed with the Incucyte ZOOM (2016A) software based on phase contrast image.

**Glucose uptake assay**. Glucose uptake was assayed according to the established protocol from a commercial glucose uptake kit (ab136955; Abcam). In brief, MDA-MB-436 cells were seeded in six-well plates followed by overnight incubation. The next day cells were treated with DMSO or indicated concentration of BAY-876. After 5 days, cells were washed three times with cold phosphate-buffered saline (PBS) and lysed with extraction buffer, frozen at −80 °C for 10 min and heated at 85 °C for 40 min. After cooling on ice for 5 min, the lysates were neutralized by adding neutralization buffer and centrifuged. The remaining lysate was then diluted with assay buffer. Finally, the colorimetric endproduct generation was set

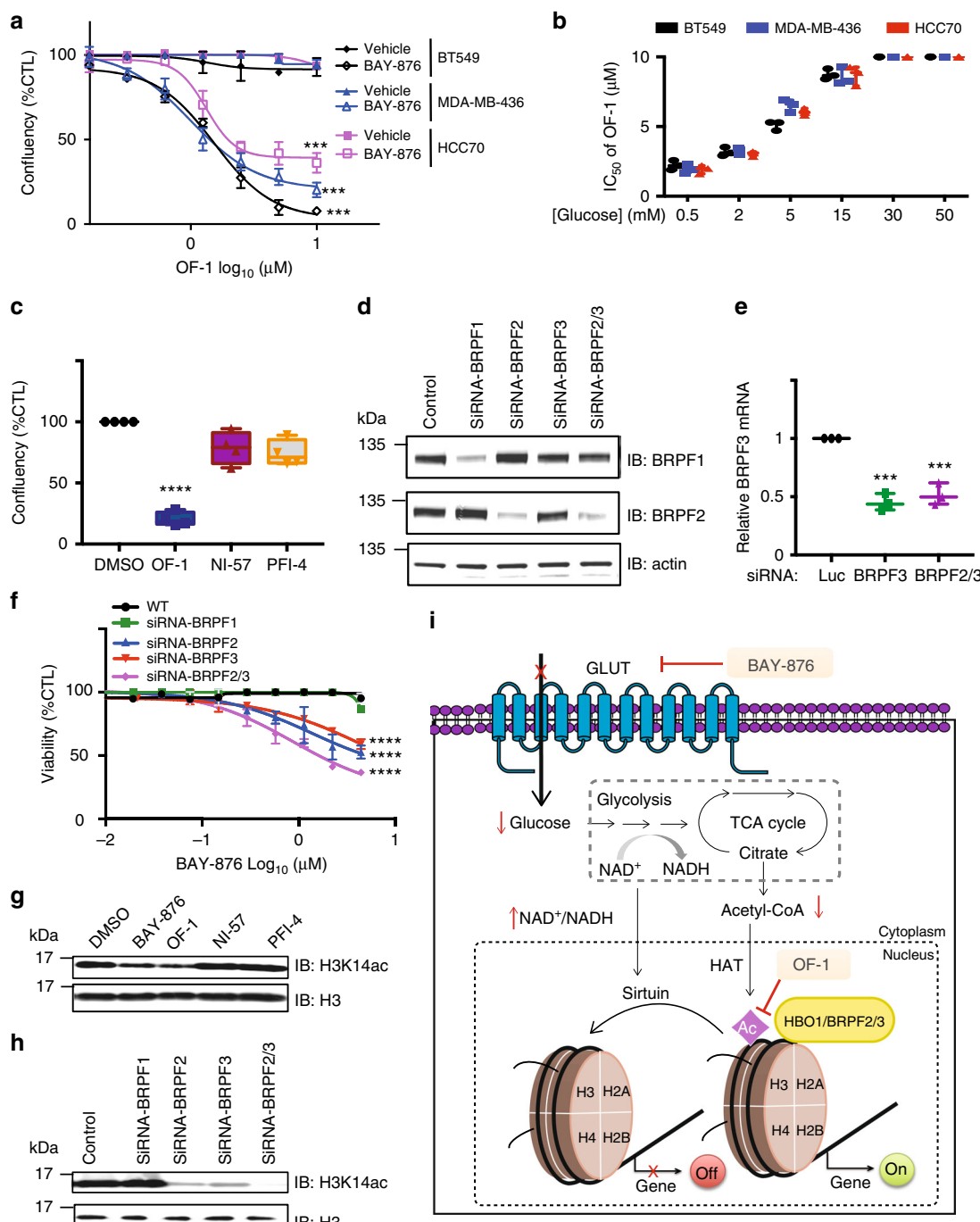

**Fig. 7** BRD inhibitors reveal a metabolic/epigenetic circuit involving HBO1 in TNBC. **a** Dose-dependent curves for cells lines treated with indicated concentrations of OF-1 with or without 3 μM BAY-876 for 7 days. Graph indicates mean, error bars denote s.d. from eight wells (from two independent assays) and p-value was computed using the Benjamini-Hochberg t test; ***$p < 0.001$. **b** Average $IC_{50}$ values of OF-1 in cells cultured under a range of glucose concentrations for the indicated three TNBC cell lines. Cells were treated with increasing doses of OF-1 for 7 days and the number of viable cells was determined by Incucyte ZOOM live cell imaging device. Error bars denote the s.d. values of independent experiments. **c** Confluency of MDA-MB-436 cells treated with indicated 3 μM BRPF inhibitors: OF-1, NI-57, and PFI-4 for 7 days. Graph indicates mean, error bars denote s.d. from eight wells (from two independent assays) and p-value was computed using the Benjamini-Hochberg t test; ***$p < 0.001$. **d** Immunoblot validation of BRPF knockdown in MDA-MB-436 cells. **e** Quantitative real-time PCR (RT-qPCR) validation of BRPF knockdown in MDA-MB-436 cells. P-value computed using Benjamini-Hochberg t test; ***$p < 0.001$. **f** Dose-dependent response of BAY-876 in BRPF-knockdown cell lines. **g** Immunoblot analysis of H3K14 acetylation in MDA-MB-436 cells following BAY-876 and BRPF inhibitor treatment for 5 days. **h** Immunoblot analysis of H3K14 acetylation in MDA-MB-436-knockdown lines. **i** Schematic illustration of a metabolic/epigenetic circuit involving GLUT1 and HBO1. Raw data images are available in Supplementary Fig. 4

up by two amplification steps according to the manufacturer's instructions in the kit and then detected at 412 nm using iMark microplate reader (Bio-Rad).

**Acetyl-CoA and NAD+/NADH ratio measurement assay**. Cells were seeded in six-well plates followed by overnight incubation. The next day, cells were treated with DMSO or the indicated concentration of BAY-876. After 5 days, cells were lysed, and intracellular metabolites were measured using commercial kits detecting acetyl-CoA (PicoProbe, Abcam ab87546) and NAD+/NADH ratio (Abcam 65348).

**Immunoblot**. Total cell lysates were resolved in 4−12% Bis-Tris protein gels (Invitrogen) with MOPS (3-(N-morpholino)propanesulfonic acid) buffer (Invitrogen) and transferred for 1.5 h (80 V) onto polyvinylidene fluoride membrane (Millipore) in Tris-glycine transfer buffer containing 20% MeOH and 0.05% sodium dodecyl sulfate. Blots were blocked for 1 h in blocking buffer (5% milk in 0.1% Tween-20 PBS) and incubated with primary antibodies: anti-ac-H3 (1:2000; ab47915); anti-ac-H4 (1:2000; ab177790); anti-H3 (1:2000; ab1791); anti-H4 (1:2000; ab10158); anti-H3K14ac (1:4000; ab52946); anit-BRPF1 (1:1000; Thermo Fisher); anti-BRPF2 (1:2000; ab71877) in blocking buffer overnight at 4 °C. After five washes with 0.1% Tween-20 PBS the blots were incubated with goat-anti-rabbit (IR800 conjugated, LiCor no. 926-32211) and donkey anti-mouse (IR 680, LiCor no. 926-68072) antibodies (1:5000) in Odyssey blocking buffer (LiCor) for 1 h at room temperature and washed five times with 0.1% Tween-20 PBS. The signal was read on an Odyssey scanner (LiCor) at 800 and 700 nm.

**Cell apoptosis assay**. Cells were seeded into 96-well plates at 3000 cells per well and left to adhere overnight. Then, cells were treated with DMSO or BAY-876 for 5 days, followed by addition of Incucyte TM Caspase- 3/7 reagents. Imaging of plates was carried out in an Incucyte Zoom instrument with a ×10 objective using the standard scan type. Data were analyzed using the integrated software provided with the instrument.

**RNA interference**. MDA-MB-436 cells were transfected with siRNA against BRPF1, BRPF2, and BRPF3. A non-targeting siRNA was used as a negative control. The Dharmacon ON-TARGET plus SMART pool Human siRNAs (25 nM final concentration) (Fisher Scientific Life Science Research, Pittsburgh, PA, USA) used for gene knockdown. The transfection protocol was performed according to the manufacturer's instructions. Seventy-two hours after transfection of knockdown efficiency was assessed by real-time PCR and Western blotting from the lysate of siRNA-transfected cells.

**Quantitative real-time PCR**. Total cellular RNA was isolated using the QIA Shredder and RNeasy kit (Qiagen, Valencia, CA, USA), as described by the manufacturer's protocol. Reverse transcription was performed using Applied Biosystems high-capacity cDNA reverse transcription kits (Invitrogen, Carlsbad, CA, USA) with random primers. To quantify gene expression, quantitative real-time PCR was performed in the Bio-Rad IQ5 system (Bio-Rad Laboratories, Hercules, CA, USA) using Finnzymes SYBR green I dye (New England Biolabs, Ipswich, MA, USA), and sequence-specific primers: BRPF3-forward: 5′-CTGGG AAGACGTGGACAACA-3′; BRPF3-reverse: 5′-TTCTGCCGAAGGGCATTGA T-3′. The 18S gene (forward: 5′- AACCCGTTGAACCCCATT-3′; reverse: 5′-CCA TCCAATCGGTAGTAGCG-3′) was used as an internal control. The reactions were performed under the following conditions: 95 °C for 3 min, followed by 45 cycles at 94 °C for 20 s, 60 °C for 30 s, and 72 °C for 20 s. The messenger RNA (mRNA) level of each gene was normalized to glyceraldehyde 3-phosphate dehydrogenase levels to obtain mRNA arbitrary units (fold change).

**Colony formation assay**. Colony formation studies were performed with TNBC cells on treatment with DMSO or the indicated concentrations of BAY-876. Cells were seeded into 6-well plates at 200 cells per well and left to adhere overnight. Then, cells were treated with DMSO or 3 μM BAY-876. After 2 weeks, the cells were stained with crystal violet and imaged.

**Muscle cell differentiation assay**. C2C12 mouse myoblast cells (ATCC CRL-1772TM) were grown in DMEM containing 20% FBS. For differentiation experiments in the presence and absence of the specified compounds, $0.3 \times 10^6$ cells were seeded and allowed to grow for 36 h before switching to differentiation media (DMEM containing 2% horse serum, 10 μgml−1 insulin, and 10 μgml−1 transferrin). Compounds were added at the time of differentiation at concentrations identical to that used for the microarray study. Cells were allowed to differentiate for 48 h before they were processed for immuofluorescence staining with α-myosin heavy-chain antibody (Red) anti-MYH1E (1:10; DSHB Hybridoma Product MF 20). MF 20 was deposited to the DSHB by Fischman, D.A. Nuclei are stained blue with DAPI (4′,6-diamidino-2-phenylindole). Post immuo-fluorescence, the differentiation index was calculated by dividing the number of nuclei in myosin heavy-chain-expressing myotubes by the total number of nuclei per field.

**RNA extraction for gene-expression analysis**. C2C12 myoblast cells that were previously cultured in growth media were switched to differentiation media and incubated with indicated compounds at concentrations of 500 nM (for JQ1 and LP-99) and 2 μM (for GSK2801, BAZ2-ICR, and BSP) for 12 h before they were harvested for extraction of total RNA. Total RNA was extracted using the PureLinkTM RNA mini kit (Thermo Fisher Scientific, 12183018A) as per the manufacturer's protocol.

**Gene-expression analysis**. RNA samples were processed for oligonucleotide microarray profiling utilizing the Affymetrix ClariomTM S Assay HT mouse. Quality controls were carried out in R (v.3.5.1) and Bioconductor[71] (v.3.7) using the arrayQualityMetrics package (v.3.36.0) taking into account array intensity distributions, distance between arrays, and variance mean dependence[72]. Principal component analysis was used to decide which arrays to process together. After quality control, arrays were corrected for background, normalized, and log2-transformed together using the rma function of the *oligo* package (v.1.44)[73] in R/Bioconductor. Probes with small variance across samples were filtered out using the genefilter (v.1.62) package, employing the nsFilter function and an interquartile range cut-off of 0.28 (or >log(1.2), determined by the midpoint of the shortest interval containing half of the data). This reduced the number of probes of the ClariomTM S array from 29,129 to 14,538.

Differential expression analysis was conducted using the limma package (v.3.36.2)[74], employing a linear model followed by empirical Bayesian analysis to determine differential expression between not-treated and treated samples. Genes were considered differentially expressed if the adjusted P-value, calculated using the Benjamini-Hochberg method in order to minimize false discovery rate (FDR), was <0.05 and the mean level of expression was >1.5-fold[75].

Gene ontology analysis was performed with the DAVID web server[76] using 1980 or 1718 genes (from the JQ1- and BSP-treated samples respectively, with FC > 1.5 and P < 0.05) against the background of 14,486 unique genes expressed in the array. Gene ontologies from all BPs were considered and those that had an arbitrary chosen FDR $<10^{-3}$ were taken into account. GSEA was performed with the Broad GSEA suite (v.2.2.4) on the collections of 4738 curated gene sets (c2), 836 transcription factors (c3), and 50 hallmarks (h) from MSigDB (v.6.1)[77]. Human gene symbols were recovered from the mouse Affymetrix ClariomTM S HT CHIP using the Chip2Chip GSEA conversion tool. Gene sets with <15 genes or more than 500 genes were excluded from the analysis, while gene sets with an FDR ≤0.25 and a nominal P ≤0.05 were considered significant. Gene ranking was performed in the weighted enrichment score using the two-sided signal-to-noise ratio and P-values were calculated using 1000 permutations of each gene set.

**Methyl transferase selectivity assays**. Assays measuring lysine and arginine methyl transferase activity have been described in Scheer et al. [78]. A radioactivity-based scintillation proximity assay was used to monitor incorporation of a tritium-labeled methyl group into the biotinylated substrate of the methyl transferase. A 10-μl reaction containing 3H-SAM and substrate at concentrations close to the apparent $K_m$ values for each enzyme was prepared. The reactions were quenched with 10 μl of 7.5 M guanidine hydrochloride; 180 μl of 20 mM Tris buffer (pH 8.0) were added; and the mixture was transferred to a 96-well FlashPlate and incubated for 1 h. The counts per minute (CPM) was measured on a TopCount plate reader. The CPM in the absence of compound or enzyme was defined as 100% activity and background (0%), respectively, for each dataset.

**Temperature shift assays**. Temperature shift assays have been described by Fedorov et al.[46]. Recombinant protein domains (bromodomain or kinases) at a concentration of 2 μM in 10 mM HEPES, pH 7.5, and 500 mM NaCl were mixed with 10 μM of chemical probes. Temperature-dependent protein unfolding profiles were measured using a Real-Time PCR Mx3005p machine (Stratagene).

**Reporting Summary**. Further information on experimental design is available in the Nature Research Reporting Summary linked to this article.

## Data availability

The data supporting the findings of this study are available within the paper and its supplementary information files and are available from the corresponding authors. Gene-expression data have been deposited in NCBI's Gene Expression Omnibus and are accessible through GEO Series accession number GSE117612.

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

## Acknowledgements

We are grateful for support by the SGC, a registered charity (number 1097737) that receives funds from AbbVie, Bayer Pharma AG, Boehringer Ingelheim, Canada Foundation for Innovation, Eshelman Institute for Innovation, Genome Canada, Innovative Medicines Initiative (EU/EFPIA) [ULTRA-DD grant no. 115766], Janssen, Merck KGaA Darmstadt Germany, MSD, Novartis Pharma AG, Ontario Ministry of Economic Development and Innovation, Pfizer, São Paulo Research Foundation-FAPESP, Takeda, and Wellcome [106169/ZZ14/Z] and the DFG funded center of excellence (CEF) at Frankfurt University. We thank the High-Throughput Genomics Group at the Oxford University for the generation of Gene Expression data, and the Terry Fox Foundation (M.L. and C.H.A.), the Canadian Institutes of Health Research (FRN-125792 to C.H.A.) and the Medical Research Council (MRC grant MR/N010051/1 to P.F.) for funding.

## Author contributions

Q.W., D.H., S.Z., A.K., K.N., S.D., E.L.-F., G.D., S.D., R.N.V., M.V., O.F., P.F., and F.L. performed experiments; S.K., Q.W., C.H.A., S.M., and S.A. wrote the manuscript, and all authors approved and edited the manuscript; P.E.B., C.H.A., S.M., S.K., M.L., J.D., M.L., and M.V. supervised research.
