## [Peer Review File · Nature Communications]

Reviewers' comments:

Reviewer #1 (Remarks to the Author):

In this manuscript, the authors present a bromodomain (BRD) centric selectivity profile of several published BRD inhibitors using 3 different biochemical assays including a commercially available ligand binding assay. They compared few BRD inhibitors in a cultured muscle cell differentiation assay and showed that only BET inhibitors can block differentiation. As last, they presented evidence that the combination of a dual BRPF2/3 inhibitor with a glucose uptake inhibitor is synergistic in inhibiting growth of TNBC cultured cells.

1) Albeit it is valuable to compare in the same assay system a panel of compounds directed against the same target class, in particular for this set of compounds, the data presented here do not add substantial new information for most of the selectivity signature that were measured in the original publications for these compounds (referenced in this ms). To strengthen the novelty of the paper, justify the claims of the authors and justify publication in this journal, it will be crucial to determine the general selectivity of this probes, i.e. outside the BRD family of proteins. This data will also be critical for the unbiased evaluation of the phenotypic effects of these compounds in cells.

2) JQ1 driven inhibition of C2C12 differentiation has been already published (Thomas C. Roberts, Usue Etxaniz, Alessandra Dall'Agnese, Shwu-Yuan Wu, Cheng-Ming Chiang, Paul E. Brennan, Matthew J. A. Wood & Pier Lorenzo Puri Scientific Reports volume 7, Article number: 6153 (2017)) as well as the role of the BET proteins in muscle differentiation (Roberts et al, Scientific reports 2017: BRD3 and BRD4 BET Bromodomain Proteins Differentially Regulate Skeletal Myogenesis). To add sufficient novelty and justify publication in this journal a more physiologically relevant model system for muscle differentiation should be used (i.e. as iPSC) for this experiment.

3) Interesting, well developed and novel is the synergy of BRPF2/3 inhibition with decrease in glucose levels for TNBC cell lines. Why is NI57 which inhibits BRPF2/3, not synergistic with BAY-876?

Additional minor comment:

The writing in the figure is often not readable (very low resolution)

Suppl Fig 2: check format: adjacent columns appear on subsequent pages

Fig 7b: Y axis : cell viability should be specified at least in figure legend

No description of glucose deprivation assay in the methods

No potency/selectivity determination for JQ1 which renders it difficult to formally exclude other target dependent effects.

No indication of concentration of compounds used in the TNBC viability assay alone or in combination with the GLUT1 inhibitor

Recommendation: publish after the suggested experiments in points 1) and 2) (or alternative experiments addressing the same criticism) are performed.

Reviewer #2 (Remarks to the Author):

In this study, Wu and colleagues report on characterization and possible applications of a set of 25 inhibitors for bromodomain. The authors evaluated selectivities of these inhibitors by measuring Kds for the interactions with a large number of bromodomains using ITC and BROMOScan. The authors further showed that the BET family selective inhibitor JQ1 but not other BD-inhibitors impedes myoblast cell differentiation and that OF-1, a selective inhibitor for BDs of BRPFs (when used in combination with a selective inhibitor for the GLUT1 transporter) displays an anti-proliferative effect in triple negative breast cancer cell lines.

This manuscript came from the team which has published the most comprehensive, remarkable study on meticulous characterization of acetyllysine binding activities of the entire family of BDs. The Cell resources 2012 paper is one of those sources that I use very often and is invaluable to the epigenetics community. The current manuscript has potential to become a valuable resource as well- it contains a set of 626 Kd values and shows an exciting example of applicability of selective inhibitors beyond JQ1. I have only a few comments primarily on description of results that could be of help in further strengthening this manuscript.

The first section of results "A set of highly selective BD probes" reads as a review and does not fit into the results section, please consider removing it entirely, or substantially shortening and moving to intro or supplement.

The results section "Selectivity of BD chemical probes" has a weak starting point/rationale. The overall paragraph, as written, highlights methods rather than the importance of knowing the precise selectivity of each inhibitor and how critical it is for BDs (for example, because it's a large family and/or the proteins are implicated in multiple sometimes orthogonal biological outcomes...)

The results section "BET chem probes block muscle..." also has a weak rationale... I would probably start with the idea that truly selective inhibitors are advantageous in exploring particular cellular events/roles of BD-containing proteins and use JQ1 here as control since its effect on myoblasts is somewhat known. Then the experiments with all other inhibitors would suggest that those BD-containing proteins do not (substantially) contribute to myoblast cell differentiation. Not sure how essential the full description of gene up/downregulation and GO analysis here, may be shorten it.

The results section "BRD chem probes reveal...", first paragraph. It may be better to describe these data in context of the second robust and important example of employing a very selective inhibitor, JQ1.

The second paragraph of this section, starting with Metabolism... needs to be developed into a separate section or two. From my point of view, this is the most exciting and novel data that should be highlighted up front. I would rewrite this part (pages 12-13) to focus on OF1 right at the beginning... currently, the long explanation of another inhibitor (BAY-876) for a glucose transporter is destructing and confusing.

References:

refs 1,2- I would certainly have the most relevant/appropriate refs here, perhaps the work of Allis lab.

for BDs in general- please add the discovery of acetyllysine binding by BDs by Ming-Ming Zhou lab.

ref. 80,81- please cite here the actual work/discoveries from Jacques Cote and XJ Yang (ref 80 is not the one from XJ's group that shows the composition of the complexes).

Page 15, refs 82-87, please add ref Andrews, 2017, PNAS that shows dual action inhibitors for BET and PI3K.

Reviewer #3 (Remarks to the Author):

Review

In this manuscript, Wu et al. present a panel of characterized 25 chemical probes with which they the role of bromodomains in different application areas, in particular in muscle cell differentiation and triple negative breast cancer.

The authors address an important problem: The function of BRDs outside of BET family is not well characterized, and few phenotypes are known for binding disruption.

Here the authors characterize a broad panel of probes and then apply this to two areas, muscle cell differentiation and TNBC. While in the first case only BET protein inhibition shows a phenotype, for TNBC the authors describe an interesting cross-talk between glucose pathways, acetylation and inhibition of BRPF proteins, involved in HAT organization. The probe OF-1 allows the authors to hone in on BRPF2 and 3, which play a role in maintaining histone H3 acetylation. Under low glucose conditions, BRPF2/3 ablation results in cell death in TNBC cells.

This paper provides a highly interesting resource and case study how to use panels of characterized probes to identify new epigenetic interactions. Together with the interesting biological findings on TNBC and BRPF2/3 this is of great interest for the community. I thus support the publication with minor corrections.

Comments:

1. The authors find OF-1 which shows distinct synergy with a GLUT1 inhibitor in TNBC cells. The mechanistic data presented in Fig. 7 is clear, but a few questions remain:
 - the authors state the PFI-4 is BRPF1 selective. This is corroborated by the binding data in the Supplementary table, however in Fig. 3C, PFI-4 shows the same binding profile as OF-1 and NI-57
 - NI-57 has an even higher affinity to BRPF3. Could the authors comment on potential origins of the lower efficacy of this compound?
2. The authors use the BromoScan technology: it would be useful to explain the method in 1-2 sentences
3. Figure 3: the panels D and E have been switched.
4. Page 8: "against all 42 BRD-containing proteins..." : in the intro, the total number of BRD-containing proteins was given as 46. What is correct?
5. Page 8: Which 15 inhibitor compounds were chosen from the panel of 25 and why those? Panel 3C should be referenced in the discussion of these experiments.
6. Figure 4C: I am not sure what the color map displays (it is only given as "high" and "low" for red to blue).
7. Page 10: The authors highlight a number of genes (MLCK, DMPK or ADAM12) which are strongly suppressed. Could the authors provide the fold change for these particular proteins (as they are not given, neither in Fig. 5A or, as far I can tell, in Fig. 5B

We are grateful to the detailed and constructive comments of the reviewers. We have now revised the manuscript considering all issues that have been raised. A detailed summary of all changes to the manuscript, new experimental data and answers to reviewer queries are outlined below.

Reviewer 1:

In this manuscript, the authors present a bromodomain (BRD) centric selectivity profile of several published BRD inhibitors using 3 different biochemical assays including a commercially available ligand binding assay. They compared few BRD inhibitors in a cultured muscle cell differentiation assay and showed that only BET inhibitors can block differentiation. As last, they presented evidence that the combination of a dual BRPF2/3 inhibitor with a glucose uptake inhibitor is synergistic in inhibiting growth of TNBC cultured cells.

1) Albeit it is valuable to compare in the same assay system a panel of compounds directed against the same target class, in particular for this set of compounds, the data presented here do not add substantial new information for most of the selectivity signature that were measured in the original publications for these compounds (referenced in this ms). To strengthen the novelty of the paper, justify the claims of the authors and justify publication in this journal, it will be crucial to determine the general selectivity of this probes, i.e. outside the BRD family of proteins. This data will also be critical for the unbiased evaluation of the phenotypic effects of these compounds in cells.

Response: While we agree with the reviewer that selectivity data of all probes within the Bromodomain have already been published, almost all selectivity data are based on T_m assays only. This assay is quite robust but particularly for stable and small protein domains it can be quite misleading as small T_m differences may indicate higher than expected binding affinities. In this paper we included therefore BromoScan data which measure direct binding constants. Correlation with T_m data is illustrated in figure 3D on the example of the panBRD inhibitor bromosporine and the excellent correlation with affinities measured using ITC is shown in 3E. We think therefore that the assessment of inhibitor selectivity within the BRD family has been significantly improved by developing this platform. Using this assay system, we included in this paper unpublished data for 15 chemical probes across 42 BRD-containing proteins with more than 600 dose response titrations (**Supplemental Table S2**). As off-target activities can be much better assessed using these new data, we therefore believe that we added a substantial amount of new data with regard to selectivity within the BRD family.

In addition, we included data on 2 previously unpublished chemical probes as well as 3 closely related derivatives (**Supplemental Figure 1**). However, we agree with the reviewer that off target activities of chemical probes would be important to access. We therefore also include now substantial new cross-screening data of the discussed chemical probes on 26 diverse protein kinases which have been reported to be frequent “off-target” family for BRD inhibitors (**new Supplemental Table S3**) as well as selectivity data on other epigenetic modulators such as lysine methyl transferases (**new Supplemental Table S4**) and arginine methyl transferases (**new Supplemental Table S5**). The screened chemical probes were inactive in all these assays and we hope that including these additional selectivity data will demonstrate sufficient selectivity outside the bromodomain family. We refer to these data now in the manuscript (page 9) but the additional selectivity data have been included in the supplemental material of the paper due to space constraints.

2) JQ1 driven inhibition of C2C12 differentiation has been already published (Thomas C. Roberts, Usue Etxaniz, Alessandra Dall’Agnese, Shwu-Yuan Wu, Cheng-Ming Chiang, Paul E. Brennan, Matthew J. A.

Wood & Pier Lorenzo Puri Scientific Reports volume 7, Article number: 6153 (2017)) as well as the role of the BET proteins in muscle differentiation (Roberts et al, Scientific reports 2017: BRD3 and BRD4 BET Bromodomain Proteins Differentially Regulate Skeletal Myogenesis). To add sufficient novelty and justify publication in this journal a more physiologically relevant model system for muscle differentiation should be used (i.e. as iPSC) for this experiment.

Response: We are aware of this study and the paper has been cited in our manuscript. The reason why we included this study was: a) we wanted to exemplify the use of the probe set by assessing potential roles of other bromodomain protein in muscle differentiation and b) we extended the published study by providing mechanistic data in form of a gene expression array study that identified key regulators of this differentiation process that are affected by chemical inhibition of BRDs. Unfortunately, the experiments showed that only BET inhibitors affected muscle cell differentiation and gene transcription of key regulators of this process. Poly-targeting of BRDs by the broad spectrum BRD inhibitor bromosporine mirrored expression changes observed using the BET inhibitor JQ1 confirming the exclusive role of BET proteins in muscle differentiation. The manuscript primarily aims to provide a resource for scientist using chemical BRD tools and we therefore think that the experiment is a good example of the use of the set.

3) Interesting, well developed and novel is the synergy of BRPF2/3 inhibition with decrease in glucose levels for TNBC cell lines. Why is NI57 which inhibits BRPF2/3, not synergistic with BAY-876?

Response: It's likely due to the solubility and cell permeability issues of NI-57. By comparing the pharmacokinetic prosperities of these two compounds, we found that OF-1 has 20-fold greater solubility and 3-fold better predicted permeability over NI-57. We mention now this limitation in the paper. Please see the results in Figure S3G and page 14 in manuscript.

Additional minor comment: The writing in the figure is often not readable (very low resolution) Suppl Fig 3: check format: adjacent columns appear on subsequent pages

Response: Thanks for flagging this problem – we revised the figures and displayed now the text using a larger font.

Fig 7b: Y axis: cell viability should be specified at least in figure legend

Response: A short method description for cell viability testing has been added in figure legend 7b in page 21 as following: “Average IC₅₀ values of OF-1 in cells cultured under a range of glucose concentrations for the indicated four TNBC lines. Cells were treated with increasing doses of OF-1 for 7 days and the number of viable cells was determined by Incucyte ZOOM live cell imaging device. Error bars denote the SDs of independent experiments.”

No description of glucose deprivation assay in the methods

Response: The glucose deprivation assay has been added in the Supplementary Materials and Methods as following:

Determination of OF-1 IC₅₀ in glucose deprivation media: Cells were seeded in 384-well plate at 500 cells per well in 40 µl media containing a range of 2X glucose concentrations for each of the indicated 4 TNBC lines. The next day, 10-point doubling dilutions of OF-1 compound starting at 20 µM were performed in glucose free media with 40 µl of each dilution subsequently added in quadruplicate to the wells of 384-well plates for final drug concentrations ranging from 10 µM to 0.02 µM in 0.1% DMSO (vol/vol). Negative control (Vehicle only) wells contained equal volumes

with 0.1% DMSO (vol/vol). Cell viability at 7 days was evaluated using Incucyte ZOOM live cell imaging device (Essen Bioscience) and analyzed with Incucyte ZOOM (2016A) software based on phase contrast image.

No potency/selectivity determination for JQ1 which renders it difficult to formally exclude other target dependent effects.

Response: We tested JQ1 sensitivity toward 2 TNBC lines in the presence or absence of BAY876, and no significant synergistic effect between JQ1 and BAY-876 was observed in BT-549 and MDA-MB436 cell lines. Please see figures S3 I-J.

No indication of concentration of compounds used in the TNBC viability assay alone or in combination with the GLUT1 inhibitor.

Answer: 3 μ M of BRPF inhibitors were used in TNBC viability assay, which has now been added in the manuscript and figure legend.

Reviewer #2:

In this study, Wu and colleagues report on characterization and possible applications of a set of 25 inhibitors for bromodomain. The authors evaluated selectivities of these inhibitors by measuring Kds for the interactions with a large number of bromodomains using ITC and BROMOScan. The authors further showed that the BET family selective inhibitor JQ1 but not other BD-inhibitors impedes myoblast cell differentiation and that OF-1, a selective inhibitor for BDs of BRPFs (when used in combination with a selective inhibitor for the GLUT1 transporter) displays an anti-proliferative effect in triple negative breast cancer cell lines. This manuscript came from the team which has published the most comprehensive, remarkable study on meticulous characterization of acetyllysine binding activities of the entire family of BDs. The Cell resources 2012 paper is one of those sources that I use very often and is invaluable to the epigenetics community. The current manuscript has potential to become a valuable resource as well- it contains a set of 626 Kd values and shows an exciting example of applicability of selective inhibitors beyond JQ1. I have only a few comments primarily on description of results that could be of help in further strengthening this manuscript.

Response: We thank the reviewer for these encouraging comments!

The first section of results "A set of highly selective BD probes" reads as a review and does not fit into the results section, please consider removing it entirely, or substantially shortening and moving to intro or supplement.

Response: We shortened and re-phrased this section in the revised version of the paper.

The results section "Selectivity of BD chemical probes" has a weak starting point/rationale. The overall paragraph, as written, highlights methods rather than the importance of knowing the precise selectivity of each inhibitor and how critical it is for BDs (for example, because it's a large family and/or the proteins are implicated in multiple sometimes orthogonal biological outcomes...).

Response: We agree with the reviewer that the introduction to this topic should be more comprehensive. We included therefore a short section discussing this issue. In addition, we screened now also the probe set against potential BRD inhibitor off-targets such as protein

kinases and other epigenetic modulators (See also our response to the first issue raised by reviewer 1).

The results section “BET chem probes block muscle...” also has a weak rationale... I would probably start with the idea that truly selective inhibitors are advantageous in exploring particular cellular events/roles of BD-containing proteins and use JQ1 here as control since its effect on myoblasts is somewhat known. Then the experiments with all other inhibitors would suggest that those BD-containing proteins do not (substantially) contribute to myoblast cell differentiation. Not sure how essential the full description of gene up/downregulation and GO analysis here, may be shorten it.

Response: We also introduced this section better in the revised version of the paper and made our rationale clearer why it was chosen. The main purpose of this system was to illustrate the use of the probe set on a complex cellular model system (and not so much to discover new biology as this manuscript should serve more as a resource rather than a primary research paper). We think therefore that previous publications on BET inhibitors in muscle cell differentiation are of advantage in order to see if other reader domains are also important in the process. We now specifically highlight the use of the panBRD inhibitor bromosporine, that showed similar transcription response to that of the BET selective inhibitor JQ1 confirming that at least among the targets tested, BET proteins are the only bromodomain targets that exert transcription control in this differentiation model.

The results section “BRD chem probes reveal...”, first paragraph. It may be better to describe these data in context of the second robust and important example of employing a very selective inhibitor, JQ1. The second paragraph of this section, starting with Metabolism... needs to be developed into a separate section or two. From my point of view, this is the most exciting and novel data that should be highlighted up front. I would rewrite this part (pages 12-13) to focus on OF1 right at the beginning... currently, the long explanation of another inhibitor (BAY-876) for a glucose transporter is destructing and confusing.

Response: We significantly edited this section and discuss now the effects of the panBRPF inhibitor OF1 under a new subheading. We left this section however after the general synergy study since we feel that this order is important to establish our hypothesis and explain the experimental design. To better highlight these results overall in the paper, we now make these findings more prominent in the abstract and introduction.

References: refs 1,2- I would certainly have the most relevant/appropriate refs here, perhaps the work of Allis lab. for BDs in general- please add the discovery of acetyllysine binding by BDs by Ming-Ming Zhou lab. ref. 80,81- please cite here the actual work/discoveries from Jacques Cote and XJ Yang (ref 80 is not the one from XJ's group that shows the composition of the complexes). Page 15, refs 82-87, please add ref Andrews, 2017, PNAS that shows dual action inhibitors for BET and PI3K.

Response: We thank the reviewer for suggesting these additional references. In the revised paper we included now:

1. Allis, C.D. & Jenuwein, T. The molecular hallmarks of epigenetic control. *Nat Rev Genet* **17**, 487-500 (2016).
2. Zeng, L. & Zhou, M.M. Bromodomain: an acetyl-lysine binding domain. *FEBS Lett* **513**, 124-8 (2002).
3. Dhalluin, C. et al. Structure and ligand of a histone acetyltransferase bromodomain. *Nature* **399**, 491-6 (1999).

4. Ullah, M. et al. Molecular architecture of quartet MOZ/MORF histone acetyltransferase complexes. *Mol Cell Biol* **28**, 6828-43 (2008).
5. Avvakumov, N. et al. Conserved molecular interactions within the HBO1 acetyltransferase complexes regulate cell proliferation. *Mol Cell Biol* **32**, 689-703 (2012).
6. Feng, Y. et al. BRPF3-HBO1 regulates replication origin activation and histone H3K14 acetylation. *EMBO J* **35**, 176-92 (2016).
7. Andrews FH et al. Dual-activity PI3K-BRD4 inhibitor for the orthogonal inhibition of MYC to block tumor growth and metastasis. *Proc Natl Acad Sci U S A* **114**, 1072-1080 (2017).

Reviewer #3

In this manuscript, Wu et al. present a panel of characterized 25 chemical probes with which they the role of bromodomains in different application areas, in particular in muscle cell differentiation and triple negative breast cancer. The authors address an important problem: The function of BRDs outside of BET family is not well characterized, and few phenotypes are known for binding disruption. Here the authors characterize a broad panel of probes and then apply this to two areas, muscle cell differentiation and TNBC. While in the first case only BET protein inhibition shows a phenotype, for TNBC the authors describe an interesting cross-talk between glucose pathways, acetylation and inhibition of BRPF proteins, involved in HAT organization. The probe OF-1 allows the authors to hone in on BRPF2 and 3, which play a role in maintaining histone H3 acetylation. Under low glucose conditions, BRPF2/3 ablation results in cell death in TNBC cells. This paper provides a highly interesting resource and case study how to use panels of characterized probes to identify new epigenetic interactions. Together with the interesting biological findings on TNBC and BRPF2/3 this is of great interest for the community. I thus support the publication with minor corrections.

We thank the reviewer for these very positive and supporting statements!

1. The authors find OF-1 which shows distinct synergy with a GLUT1 inhibitor in TNBC cells. The mechanistic data presented in Fig. 7 is clear, but a few questions remain: - the authors state the PFI-4 is BRPF1 selective. This is corroborated by the binding data in the Supplementary table, however in Fig. 3C, PFI-4 shows the same binding profile as OF-1 and Ni-57

Response: The colouring in this figure is somewhat misleading due to the rather large binning steps. But BRPF3 colouring was also wrong and we corrected this in the revised version. PFI-4 has a Kd of 5,4 nM for BRPF1, 270 nM for BRPF2 (BRD1) and 1700 nM for BRPF3. Thus, this probe is about 50 fold selective against BRPF2 and 300 fold selective against BRPF3.

- NI-57 has an even higher affinity to BRPF3. Could the authors comment on potential origins of the lower efficacy of this compound?

Response: It's likely due to the solubility and cell permeability issues of NI-57. Please see the results in Figure S2G. We discuss this issue now on page 15 in the revised manuscript.

2. The authors use the BromoScan technology: it would be useful to explain the method in 1-2 sentences.

Response: We introduce this technology now on page 8 of the revised manuscript.

3. Figure 3: the panels D and E have been switched.

Response: Thanks for pointing this out. We corrected the figure.

4. Page 8: “against all 42 BRD-containing proteins...” : in the intro, the total number of BRD-containing proteins was given as 46. What is correct?

Response: 42 refers to the number of screening bromodomains in the BROMOScan panel and not to the number of total BRD-containing proteins. We rephrased this sentence to make this clearer.

5. Page 8: Which 15 inhibitor compounds were chosen from the panel of 25 and why those? Panel 3C should be referenced in the discussion of these experiments.

Response: We now refer to figure 3C in the paper. Many of the experiments were carried out over several years as part of larger screening campaigns including also the cell based assays. Since more inhibitors are constantly developed we refer the reader now to our web-based database (<https://www.thesgc.org/chemical-probes>) which allows us to update characterization data and add new compounds to the set.

6. Figure 4C: I am not sure what the color map displays (it is only given as “high” and “low” for red to blue).

Response: We labelled now the figure capture describing the level of expression changes.

7. Page 10: The authors highlight a number of genes (MLCK, DMPK or ADAM12) which are strongly suppressed. Could the authors provide the fold change for these particular proteins (as they are not given, neither in Fig. 5A or, as far I can tell, in Fig. 5B

Response: This information has now been included. We show significantly regulated genes now in the volcano blot (Figure 5).

REVIEWERS' COMMENTS:

Reviewer #1 (Remarks to the Author):

Since the authors have satisfactorily responded to the questions raised there are no objections to proceed with publication in Nat communication. We recommend addition of adequate legends to the newly added Supp Figures (i.e. Supp Table 3)

Reviewer #2 (Remarks to the Author):

The authors have addressed all my previous comments. with best regards, Tatiana Kutateladze

Reviewer #3 (Remarks to the Author):

All my comments have been addressed and I thus support publication of this study.

Reviewer #1 (Remarks to the Author):

We recommend addition of adequate legends to the newly added Supp Figures (i.e. Supp Table 3)

We included now legends to the new supplementary tables.